# Combining Log Files and Monitoring Data to Detect Anomaly Patterns in a Data Center

**Laura Viola [1]**, **Elisabetta Ronchieri [1,2],\*** and **Claudia Cavallaro [3],\***

1 Department of Statistical Sciences, University of Bologna, 40126 Bologna, Italy; laura.viola5@studio.unibo.it
2 INFN CNAF, 40126 Bologna, Italy
3 Department of Mathematics and Computer Science, University of Catania, 95124 Catania, Italy
* Correspondence: elisabetta.ronchieri@cnaf.infn.it (E.R.); claudia.cavallaro@unict.it (C.C.);
  Tel.: +39-0512095072 (E.R.)

**Abstract:** Context—Anomaly detection in a data center is a challenging task, having to consider different services on various resources. Current literature shows the application of artificial intelligence and machine learning techniques to either log files or monitoring data: the former created by services at run time, while the latter produced by specific sensors directly on the physical or virtual machine. Objectives—We propose a model that exploits information both in log files and monitoring data to identify patterns and detect anomalies over time both at the service level and at the machine level. Methods—The key idea is to construct a specific dictionary for each log file which helps to extract anomalous *n*-grams in the feature matrix. Several techniques of Natural Language Processing, such as wordclouds and Topic modeling, have been used to enrich such dictionary. A clustering algorithm was then applied to the feature matrix to identify and group the various types of anomalies. On the other side, time series anomaly detection technique has been applied to sensors data in order to combine problems found in the log files with problems stored in the monitoring data. Several services (i.e., log files) running on the same machine have been grouped together with the monitoring metrics. Results—We have tested our approach on a real data center equipped with log files and monitoring data that can characterize the behaviour of physical and virtual resources in production. The data have been provided by the National Institute for Nuclear Physics in Italy. We have observed a correspondence between anomalies in log files and monitoring data, e.g., a decrease in memory usage or an increase in machine load. The results are extremely promising. Conclusions—Important outcomes have emerged thanks to the integration between these two types of data. Our model requires to integrate site administrators' expertise in order to consider all critical scenarios in the data center and understand results properly.

**Keywords:** log analysis; monitoring data; anomaly detection; natural language processing; topic modeling; clustering technique; time series anomaly detection

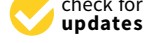



## 1. Introduction

A data center usually handles a huge amount of different resources, where a plethora of services can run, collecting information about computer systems and machine states. It is quite common to have services that store their events in specific files, known as log files, allowing site administrators to understand their current functioning and react proactively in case of issues. Furthermore, data center monitors the physical and virtual machines, tracing different devices readings (such as CPU load, memory, disk space and so on), information about network traffic and bandwidth usage, in order to send alerts to administrators if critical situations happen. Services produce log files, while monitoring sensors measure different metrics with respect to the resource level they refer to.

Logs are created by a service and contain semi-structured texts that are appended to a file with the *.log* extension. These files grow in size and can become very large, but



they tend to include similar texts over time that may cover various aspects of each service, such as warnings, errors, activity information and so on. This leads to develop solutions that automate the processing of logs and support system administrators while analyzing systems' health. Log files usually do not contain the same type of information, e.g.,: *syslog* event logs a system activity, *crond* event logs the CRON entries that show up in *syslog*, and *virtlog* event logs virtualization-related operations. Each file tends to describe a partial view of the whole machine. The stored information can contain: the time and date of specific event to log exactly what happened; the process name and process identifier; the machine hostname and its internet protocol address. The services can use different keywords to express normal or erroneous behaviour.

Monitoring data are created by sensors that run on the machine, containing e.g., a timestamp expressed in epoch time, metric's name, metric's value, and the machine hostname. These measurements cover all the running state of the machine, therefore site administrators can decide to store them with different frequency. An alarming email can be sent to the site administrators when a data center-based threshold is exceeded, such as when the used memory of a machine has reached 95% of the total memory.

Some studies have proposed approaches to handle the error problems from manual operation to automated operation [1]. They define pipelines that include the transformation of log files into a more readable format understandable by analysis tools, such as *.csv* and *.json*, the classification of observations with respect to the threshold values, and the extraction of any suspicious information, like the *anomaly* term. Quite often it is not feasible just selecting the file and retyping the file extension as *.csv* or *.json*, because the transformed file could be wrongly reformatted, e.g., containing multiple rows of heading variables. Furthermore, the file usually contains daily service information, so the number of data can be over 60–120 K rows, penalizing the usage of some analysis tools. Before starting any analysis, it is essential to decide which variables have to be included in the resultant data sets. Spreading the data across multiple columns is another aspect to consider in order to organise your data set into a manageable format. The resultant files can be used to identify trends and unusual activities that are beneficial for both short- and long-term data center management.

In our previous work [2] we have just considered log files to identify anomaly detection patterns by using natural language processing (NLP), autoencoder and invariant mining techniques. In this new study we have combined the knowledge present in monitoring information and log files in order to improve our analysis. Artificial intelligence and machine learning techniques are promising solutions to automatically identify anomaly detection patterns and predict failures in a machine. Different methodologies have been developed to separately identify anomalies in the two data sources. All the phases are aimed at identifying anomalous messages in the logs. The core of our methodology lies in the creation of an anomaly dictionary for each log file. This dictionary allowed us to generate a feature matrix closely related to the semantic areas of anomalies in that particular service. Log files contain a considerable amount of texts, therefore NLP methods, such as word-clouds, topic modeling, exploring the list of unique *n*-grams, are needed to enrich the dictionary. Starting from the matrix of the features created, it will then be possible to apply a clustering algorithm to not only distinguish anomalous messages from non-anomalous ones, but also to differentiate the different types of anomalous events. The proposed approach is repeatable in other contexts and domains. With minimum setup effort and the usage of artificial intelligence and statistical tools, it is possible to create an effective anomaly dictionary. For monitoring data we decided to apply JumpStarter technique [3] in order to obtain the anomaly score associated with each temporal instances based on multiple time series. We have decided to aggregate the two data sources at machine level by considering all the log files for the running services and the measurements of machine sensors in order to determine the variation of values nearby an anomaly.

Through experiments, we illustrate the potential benefits of our approach by answering our research questions that can be summarized as follows:

**RQ1.** Can we get service anomalies by considering log messages?
**RQ2.** Are there NLP techniques that can automatically provide information on the state of a service?
**RQ3.** Can we get machine's state by looking at monitoring metrics' data?
**RQ4.** Can we relate log and monitoring data to determine anomalous behaviour at machine level?

The remainder of this paper is as follows. Section 2 talks about related work. Section 3 provides information about data that we have considered for this study. Then, section 4 introduces the rationale of our approach. Section 5 talks about anomaly detection methodologies used for log files, while Section 6 talks about those for monitoring metrics. Section 7 provides some results at machine level by combining the two types of data. Section 8 concludes the paper discussing the presented work.

## 2. Related Works

The existing literature on anomaly detection is vast. The main approaches both exploit NLP techniques on top of log files, produced by services running on a given machine, and use numerical data coming from monitoring system.

### 2.1. Log Data

Log files are extremely important and useful sources of information for a specific service. There exist different types of messages, such as INFO messages that aim at giving some knowledge about the state of a service. There are other types of messages that include warnings, alerts, fatal errors, failures, alarms, debugs. Some of these messages can be more serious than others but not blocking the current running service (e.g., warnings and debugs), while others do not allow the service to work (e.g., alerts, fatal and errors).

Log messages are string objects, therefore NLP techniques represent solutions to preprocess them. All the studies related to NLP include a preprocessing phase to extract relevant information from the data. Authors, who work with log messages, have developed their models by using Word2Vec and other techniques for the preprocessing phase. Bertero et al. [4] have developed three models—a Neural Network, a Gaussian Naive Bayes and a Random Forest Classifier—feeding them with log data vectorized through Word2Vec. Wang et al. [5] have fed their models with vectorized data determined with Word2Vec and Term Frequency-Inverse Document Frequency (TF-IDF), to develop a Gradient Boosting Decision Tree, a Naive Bayes, and a Recurrent Neural Network (a Long Short Term Memory model, LSTM) to exploit the ability of LSTM to deal with sequential data. Lukas et al. [6] have developed a Recurrent Neural Network too and they also started from Word2Vec.

All the mentioned studies have used labelled data and supervised learning solutions. However, log data are usually characterized by having imbalanced class distribution, because anomalies are not requested to appear often in the log files. TF-IDF alone is not a valuable technique for log analysis, because it does not allow to consider the relevance of a word in a message. Furthermore, in anomaly detection it is often the case that labelled data are not available and unsupervised solutions are required. Zeufack et al. [7] and Bursic et al. [8] have adopted an OPTICS clustering algorithm (density-based clustering) and an Autoencoder Neural Network respectively to solve the problem of anomalies.

If the number of clusters set in input is too large, the algorithm will return false results, failing to correctly distinguish the anomalous instances [9].

### 2.2. Monitoring Data

These data usually derive from the monitoring service that has been setup in a data center. They represent another important source of information in order to understand the health state of machines. The measure typically depends on time. Therefore time series anomaly techniques can be used to discriminate between normal and abnormal systems' behaviours.

Huang et al. [10] have taken into analysis voltage time series, current time series, and temperature time series and employed a simple effective technique aimed at identifying thresholds for normal and anomalous behaviors based on mean and variance values.

Gabel et al. [11] have adopted the Tukey method, where observations occurring around a certain central value are recognised as normal and the more they move from this central value the higher the probability they represent an anomalous event. The authors have also used some other geometry-based techniques, such as the Sign Test and the Local Outlier Factor test. Wang et al. [12] have developed an anomaly detection method based on the Tukey test, but they also developed a framework of anomaly recognition based on multivariate goodness-of-fit: a normal behavior distribution is given and compared to other time series to check for anomalies through a likelihood ratio test.

Decker et al. [13] have defined threshold starting from mean and standard deviation values. However, they have not focused on the whole time series but they have proceed by segments, called sliding windows, for a more precise computation. Moreover, they have also developed an Evolving Gaussian Fuzzy Classifier to cluster without any prior knowledge different segments, based on their distribution through an adaptive discriminative algorithm which is able to update its rules based on new observed values.

Ma et al. [3] have proposed a jumpstarting multivariate time series anomaly detection approach based on the compressed sensing technique for a short initialization time.

### 2.3. Multi-Sources Data

Literature where log data and monitoring data are combined is not much developed, in particular for anomaly detection.

Nti et al. [14] have developed a very effective model where text data and numeric data, after proper preprocessing, have been combined together to develop a model for stock price prediction. On the other hand, Lee et al. [15] have developed a model to estimate the risk of bidding projects in urban engineering by combining numerical metrics and textual reviews. However, in both of these two studies we have observed that the data fusion process was not very problematic: in the first work the availability of finance-related textual data is enough to be able to look for data in the required moment in time, while in the second study every textual information has been clearly assigned.

## 3. Source Data

Our work has started with dataset selection. In this work we have considered the same set of log files used in the previous study [2]. The log files are related to a set of services running on machines at INFN Tier-1 data center [16] that are used by the large hadron collider experiments [17]. Figure 1 shows four categories of resources which make up the TIER-1 infrastructure and cooperate with each other: Storage includes disk and tape storage services, and data transfer services; Farming handles the computational resources; Network is responsible for security rules and access to the various resources in the infrastructure; User Support handles identification and resolution of problems, authentication and authorization problems, creation of accounts and resource configuration.

Each log mainly belongs to Linux system services, such as the software utility *crond*, the free and open-source main transfer agent postfix and the standard for message logging *syslog*. Table 1 summarizes the first 30 log filenames according to their frequency, that is the number of times a value of an unique filename occurs. The overall log suffix-type frequencies is 3,562,759. However, on the Linux machine other suffix-types of files (containing log entries) are available, such as *.gz* and *.txt* respectively with 10,869 and 2 frequencies. The *.gz* files are used for large log files, while the *.txt* files are for small log files.

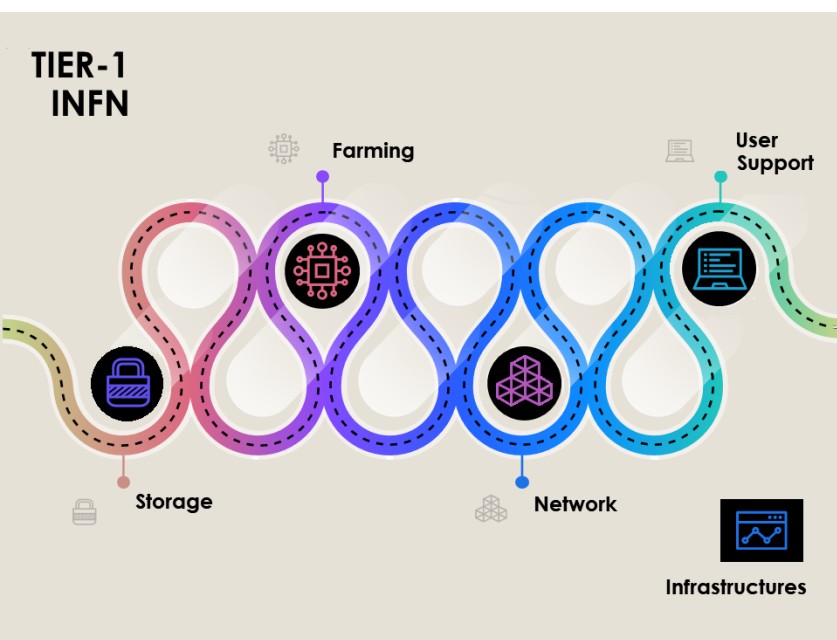

**Figure 1.** The categories of resources making up the Tier-1 infrastructure at CNAF-INFN.

**Table 1.** The top 30 log files per frequency.

| Filename | Frequency | Filename | Frequency | Filename | Frequency |
|---|---|---|---|---|---|
| sudo.log | 378,781 | systemd.log | 107,700 | userhelper.log | 21,380 |
| puppet-agent.log | 368,530 | mmfs.log | 72,620 | nslcd.log | 20,544 |
| run-parts.log | 365,734 | rsyslogd.log | 70,210 | neutron_linuxbridge.log | 8572 |
| crontab.log | 348,896 | kernel.log | 65,938 | runuser.log | 6859 |
| crond.log | 347,708 | logrotate.log | 62,531 | cvmfs_x509_validator.log | 6031 |
| sshd.log | 303,919 | syslog.log | 47,330 | cvmfs_x509_helper.log | 5399 |
| anacron.log | 287,419 | yum.log | 43,301 | srp_daemon.log | 4938 |
| postfix.log | 175,558 | fusinv-agent.log | 42,125 | edg-mkgridmap.log | 4083 |
| auditd.log | 120,473 | root.log | 37,345 | libvirtd.log | 3328 |
| smartd.log | 109,441 | gpfs.log | 31,000 | dbus.log | 3301 |

Each machine can run different services and it is usually characterized by a certain amount of memory and disk space. To check each machine status, the data center exploits a monitoring system that is able to measure various metrics, show them through the *graphana* service and store them through an *influxd* service [18].

On the basis of site administrators' feedback, in this study we have considered a subset of metrics grouped in three categories as shown in Table 2: load average that refers to the average system load on a server for a specific period of time; memory that provides information about the consumption of memory; iostat average that captures the status of the device.

Load averages are usually three numbers, showing the average load in the last minute, in the last five minutes, and in the last fifteen minutes: if the one minute average is higher than the five or fifteen minute averages, then load is increasing; if the five minute average is lower than the five or fifteen minute averages, then load is decreasing. The memory category provides information about the normal and swap memory: when the swap memory is used, then it means that the normal memory is full. The used memory is one of the memory metrics for which a warning message is automatically sent by email to site administrators when its value exceeds 95% of total memory; iostat averages give input and output devices utilization.

**Table 2.** Monitoring metrics.

| Category | Metrics | Category | Metrics | Category | Metrics |
|---|---|---|---|---|---|
| Linux load average | 1 min | Memory | swap free | iostat average | cpu pct iowait |
| Linux load average | 5 min | Memory | swap total | iostat average | cpu pct nice |
| Linux load average | 15 min | Memory | swap used | iostat average | cpu pct steal |
| | | Memory | available | iostat average | cpu pct system |
| | | Memory | buffers | iostat average | cpu pct user |
| | | Memory | cached | iostat average | cpu pct idle |
| | | Memory | dirty | | |
| | | Memory | free | | |
| | | Memory | used | | |
| | | Memory | total | | |

*3.1. Log Files*

Each of these log files contains a different amount of lines. They contain numerical and textual data that describe system states and run time information. Each log entry includes a message that contains a natural-language text (i.e., a list of words) describing some events. Logs are generally generated by logging statements inserted, either by software developers in source code or by system administrators in configuration files, to record particular events and software behaviour. Figures 2 and 3 show two different log entry samples composed of a log header and a log message: the former is generally composed of a timestamp, a custom-configuration information (such as the hostname in Figure 2, where the service runs and a log level verbosity in Figure 3) and the name of the service the message is associated to; the latter is just the message that contains information of the logged event. The log header structure is relatively different in both figures: this is one of the reasons for which the application of a non service-specific solution when parsing log files is a hard task.

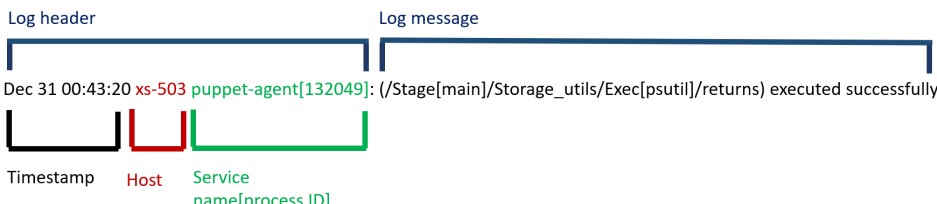

**Figure 2.** A log entry sample from the puppet-agent service.

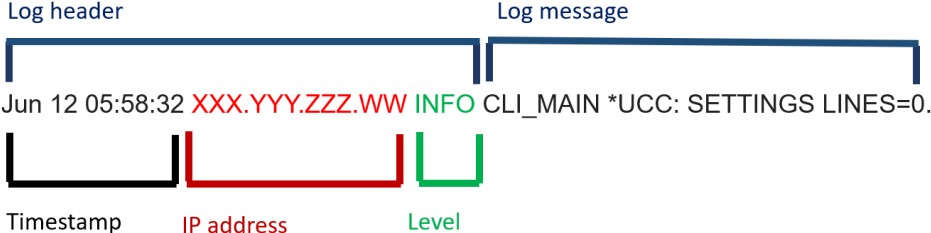

**Figure 3.** A  log entry sample from the puppet-agent service.

Figures 4 and 5 show two examples of the log message of a log entry, characterized by a natural-language text whose interpretation is difficult because there is not an official standard defining the message format. The text is usually composed of different fields called dynamic and static: a dynamic field is a string or a set of strings that are assigned at run time; a static field does not change during events. Such fields can be delimited by different separators, such as a comma, a white space or a parenthesis. Logging practice is scarcely well documented.

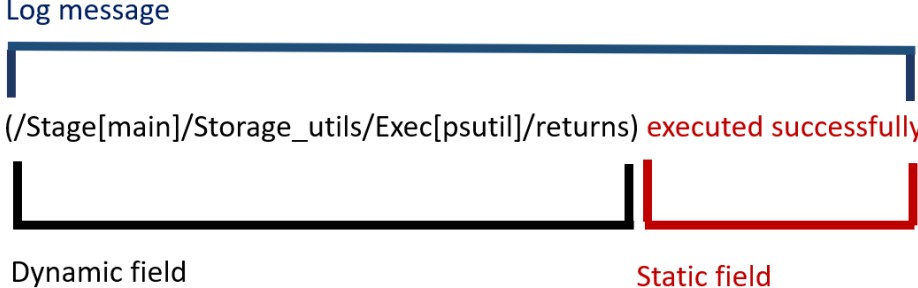

**Figure 4.** A log message fields with just one dynamic field and static field.

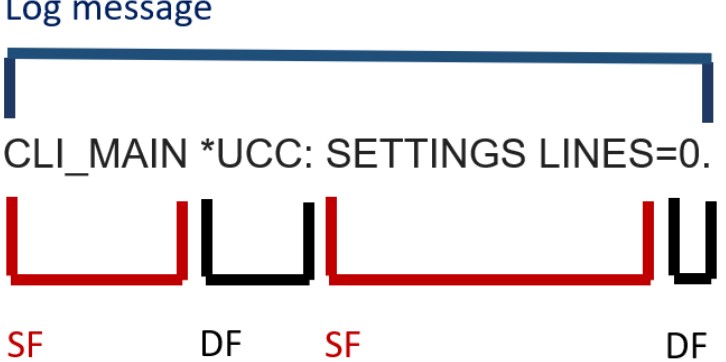

**Figure 5.** A log message fields with a sequence of dynamic and static fields.

This activity mainly depends on human expertise [19]. They often have to analyze a large volume of information that may be unrelated to the problematic scenarios and lead to overwhelming messages [20]. It's important to mention that not all log entries of a log file have the same static field. Despite this, it is extremely common to find different log messages sharing the same static field.

*3.2. Monitoring Metric Files*

Each of these files contains a different amount of lines. They contain numerical and textual data that describe machine state for a specific metric over time. Each file includes the hostname of the machine, epoch time and measurements. They are produced by using the *influx* client that queries a specific database (defined for a set of machines) to extract a given metric in a certain period of time, and dumps values into a *.csv* file. Below it is reported an example of the query performed by command line:

```
$ influx -host=<service hostname> -port=<port number> \
-username=''<user>'' -password=''<password>'' \
-database=''<database>'' \
-execute=''SELECT * FROM \''all_data\''.\''load_avg.five\'' \
WHERE time > now()---<number of~days> GROUP BY \''host\'''' \
-format=csv > one_week.csv
```

## 4. Methodology Overview

In our previous work we have applied word2vec, autoencoder and invariant mining techniques [2] to identify anomaly patterns from log files. The autoencoder was used to learn a more efficient representation of data, while minimizing the corresponding error. The invariant mining was used to collect error messages that generate a problem in a physical or virtual machine at INFN Tier-1 data center. The main objective of this study has been the development and validation of an anomaly detection model by considering a data center use case. A generalized workflow is shown in Figure 6, which schematizes the steps that have been undertaken to implement this model.

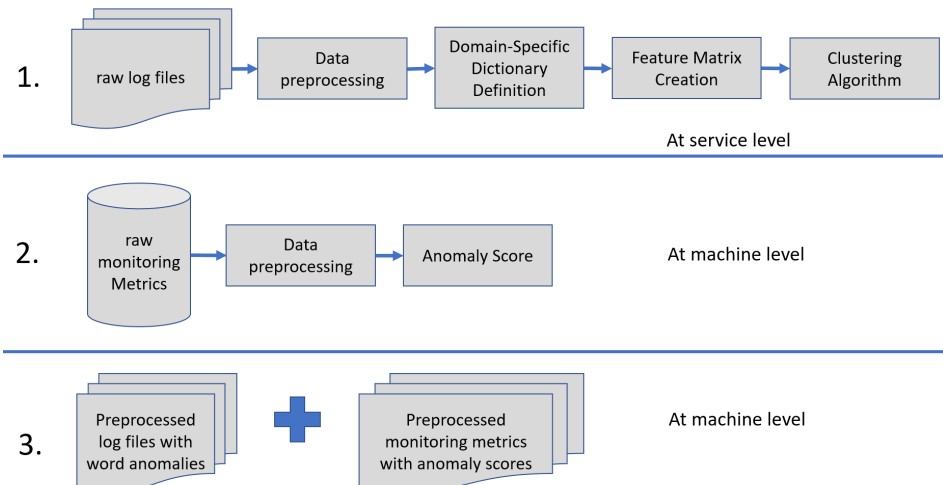

**Figure 6.** General methodology overview.

Once data has been collected both at service level and machine level respectively with a set of log files and monitoring metrics' values, we have dedicated effort to develop algorithms in order to identify anomalies in these two data sources separately.

On the two types of data we have applied different techniques. NLP techniques have been applied to log messages to extract relevant information from the data. Time series anomaly technique has been applied to monitoring data to discriminate between normal and abnormal systems' variations.

Furthermore, in our study, log data and monitoring metrics are generated by two different systems, that can cause a deep time discrepancy. Therefore, one of the main challenges, we have tried to deal with, has been to find a suitable, reliable and consistent way of putting into relation these two kind of data, avoiding the serious risk of relating data which are actually unrelated. In this study, the behavior of the metrics in correspondence with anomalies in the logs is observed graphically. Future studies will further investigate how to integrate this data into a single model.

*Project Implementation*

To implement this study we have used Python libraries and Jupyter notebook with Python version >= 3.8, uploading code, images, and results in GitLab project, that will be made available when the project will be properly documented.

To preprocess data, we have used the nltk library [21] that provides methods to clean data. We have also used ToktokTokenizer() from nltk to tokenize one sentence per line. To convert text documents to a matrix of token counts, we preferred to use CountVectorizer() [22] from sklearn as it fits better with the type of clustering algorithm used later. CountVectorizer() has been used in two ways. Firstly, we use it to determine the whole list of unique *n*-grams in every log gile. Secondly, we use CountVectorizer() to build the feature matrix by considering *n*-grams included in the dictionary: the number of times one *n*-gram value is in the message. To apply TF-IDf, we have used TfidfTransformer() from sklearn.

To apply LDA, we have used pyLDAvis [23], that is able to represent the topics in a topic model that has been fit to a corpus of text data. The package uses LDA to inform an interactive web-based visualization..

## 5. Log Anomaly Detection

The main phases that led to the identification of the log anomalies are divided as follows: data preprocessing, exploration of messages and creation of the dictionary, creation of the feature matrix and clustering algorithm.

### 5.1. Data Preprocessing

During the data preprocessing we have first changed the format of log files, which have turned into *.csv* files by applying service-specific procedure, because we have had to take into consideration different log header structures (as shown in Figures 2 and 3), one per service. This procedure is able to manage each entry of the log file, performing at least the following operations: adding e.g., either internet protocol (ip) address or hostname when they are missing, service name when it is missing; splitting service name and process identifier (id); and getting component name. When the log file is turned into *.csv* format, the entry in the *.csv* file contains the log message (msg) and a set of other variables (see Figure 7 for a graphical representation), such as: date, time, timestamp, hostname, ip address, service name, id, component name. The hostname and ip couples are not always both available, especially when the service runs on a virtual machine. Each file is related to a particular service that runs on a well-known machine in the data center. Its location is obtained by a local database and included into the resulting file.

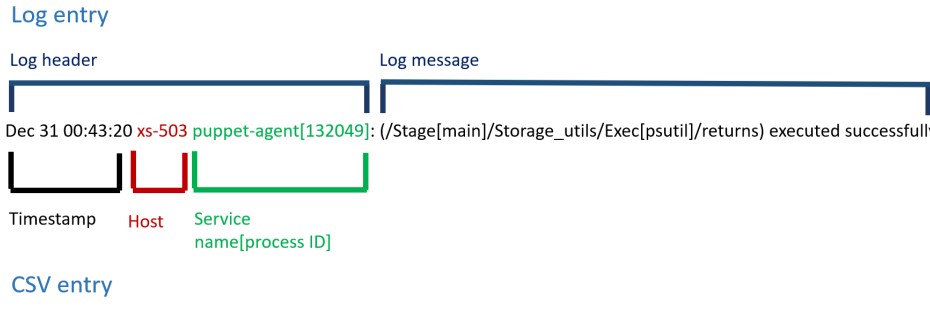

**Figure 7.** An example of log header transformed into CSV header.

In this phase we have tackled some site administration peculiarities: the same service is called either in lower or capital letters; the process identifier is included in the service logging file; the service name is included (or not) in the log message; the process identifier is included (or not) in the log message; the logging filename includes typo error. We have also identified some typo errors in the log messages. Before performing any cleaning operations, we have excluded meaningless services' log files, especially those with a small number of events. In the remaining logs, the following changes have been applied in the message field: the removal of unwanted texts, such as punctuation, non-alphanumeric characters, and any other kind of characters that are not part of the language by involving regular expressions; the exclusion of non-English characters; the stopwords removal, that is frequent general words (like *of*, *are*, *the*, *it*, *is*) with a low meaning. For stopwords, we have decided to keep negative terms, and other words that may refer to a problem, such as *up*, *again*, *too*, *ok*, *out*, *yet*, and *more*. Each letter has been converted to lowercase to avoid case-sensitivity problems.

### 5.2. Creation of the Anomaly Dictionary

The messages in the logs are characterized by a rich and technical vocabulary. Although very numerous, the number of static fields observable within a log files is limited. Until now, log anomaly detection methodologies have not used a dictionary to better filter patterns related to the semantic area of anomalies. After the cleaning step, a feature extraction techniques, such as Word2Vec and TF-IDF, is usually applied. Going into detail, TF-IDF is often used to find the important words in a collection of documents: TF stands for term frequency matrix, measuring the association of a word with respect to a given document; while IDF stands for inverse document frequency, representing the importance of one word [24]. If a word appears in many log files, the importance of this word will be decreased. This approach could not be necessarily capable of providing useful information for the identification of anomalies if applied alone, because the search for features is not strictly connected to the search for patterns belonging to the semantic area of anomalies.

In addition, the feature matrix may have high dimension since each unique $n$-gram in all messages will compose a column of such matrix. This is a problem not to be underestimated in the case of massive log files typically recorded in a data center. Each message is composed by various dynamic fields; therefore, we have used several $n$-grams with $n <= 5$ to rebuild anomalous message, omitting dynamic fields. The n value selection depends on messages included in the log files.

Our main idea is to reduce the number of columns generated, restricting only to those $n$-grams that can be associated with anomalies or that better describe an anomalous message. To do this it was decided to create a dictionary. The dictionary is nothing more than a set of words or sequences of words related to the semantic area of anomalies for that particular log file. For each log files it makes sense to build a different dictionary as the message structure and the type of anomalies are different between services. Doing so, we have been able to identify relevant terms to a global dictionary. An important contribution in the creation of the dictionary could have been made by the experts of the data center. The following anomalies have been suggested by programmers: *abort, aborted, aborting, alert, cannot, can't, couldn't, deprecated, deprecate, disabled, error, exception, fail, fails, failure, failed, failing, fatality, fatal, invalid, impossible, huped, misconfiguration, problem, sslerror, suppressed, suppressing, suspended, suspend, suspending, suspension, stopped, stopping, stop, unable, unsupported, warning, warn, warned.*

To create an effective dictionary it is necessary to extensively explore the log messages. In this phase we have started to trace the types of log events, such as abort, fail and invalid, and to identify anomaly key terms that can be used to classify the reason of the problems in the service. This part of the study has been applied to all the set of files examined contributing to a better understanding of the variation in the machine status. However, Table 3 summarizes a couple of message lines that describe a wrong service behaviour.

**Table 3.** Examples of message lines for the crond log file.

| Log Event msg | Log Event Type | Anomaly Key Term |
|---|---|---|
| .. reset error counters | error | reset |
| .. failed create session connection time out | fail | time out |

To enrich the dictionary, several NLP solutions were used. It might be useful to explore the complete list of unique $n$-grams starting from the texts and insert those who refers to anomalies in the dictionary. We have used the whole list of terms and sequences provided by $n$-grams. We have considered the least frequent, mainly because a service is built to work, therefore anomaly terms have a lower frequency. $n$-grams of medium frequency have also been considered. We have decided to consider $n \in [3, 5]$ because they are able to rebuild the meaning of each message better than with $n$ equal to 2. For simple and small log file, whenever possible we have performed a first handson check of the list of unique $n$-grams, and added them to anomaly dictionary.

Furthermore, representing the most frequent and least frequent words via word cloud can be useful for bringing out anomalous patterns in the case of large log files, especially rare words. Figure 8 shows two word cloud of the first (and last) thirty subsequences of length 3 from *audit* log messages sequences [25]. For example, in the wordcloud in Figure 8 the anomalous "suspending logging" pattern appears among the less frequent tri-grams. Analyzing the list of the various $n$-grams, we have seen that some messages containing "suspending logging" continue with "due previously mentioned write error". Consequently, in addition to "suspending logging", "previously mentioned write error" has also been added to the dictionary. This will be essential to add information and to track anomalies in order to create groups of different types of anomalies.

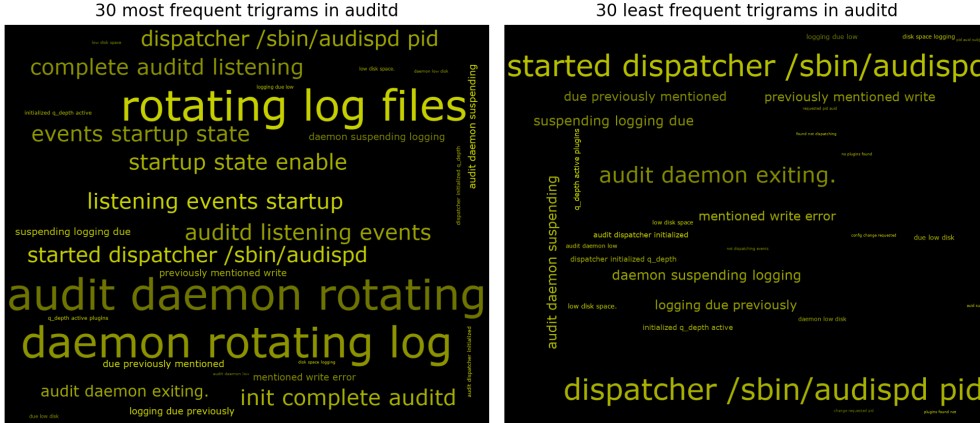

**Figure 8.** Word cloud for the *auditd* service.

Topic modeling can be another useful technique to find latent topics in mostly unstructured collections of log messages, and to identify word structures and anomalies that belong to specific topic. We do not identify any latent topic. In this study we applied Latent Dirichlet Allocation (LDA). LDA assumes that documents are a mixture of topics, while topics are a distribution of words [26]. The optimal number of topics was determined by maximizing the difference between the overall topic coherence and the average topic overlap, calculated as the mean of the 'Jaccard Similarity' values between topics. LDA has been used to deeply explore each message. In this study we have considered single terms for topic modeling due to the service specific knowledge included in each message (i.e., url, process name and so on).

Figures 9 and 10 show how terms are distributed in the different topics for the *screen* and *virtlogd* services. In both figures we can observe that during the preprocessing activity we have cleaned log messages keeping meaningful information, such as the function name, the name of the machine, users' names (that for privacy reason we have omitted), and potential problems. For example, Figure 9 Topic 2 shows words like *auth*, *failure*, *authentication*, *screen* and *pam_ldap*, that can indicate a *screen authentication failure*. We have included in the dictionary construction, anomaly patterns resulting from LDA. For example, according to Figure 9, we have definitely added words *failure* and *authentication failure* from Topic 2; *fails*, *failed authentication* from Topic 0; *unknown* from Topic 3. Instead we have omitted Topic 1 that contains *verified* and *authentication* that can indicate no problem.

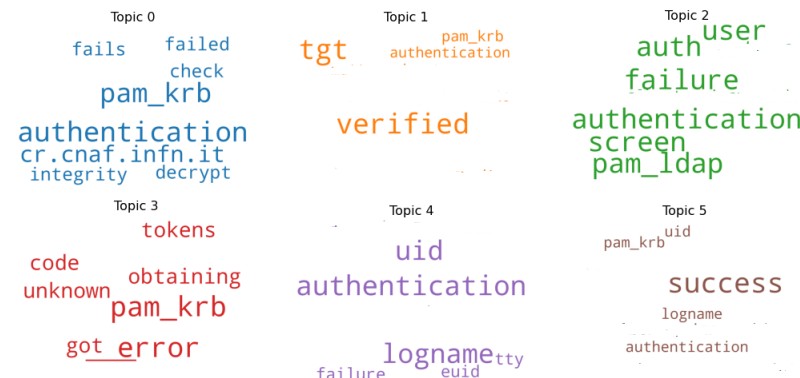

**Figure 9.** Identified topics for the *screen* service.

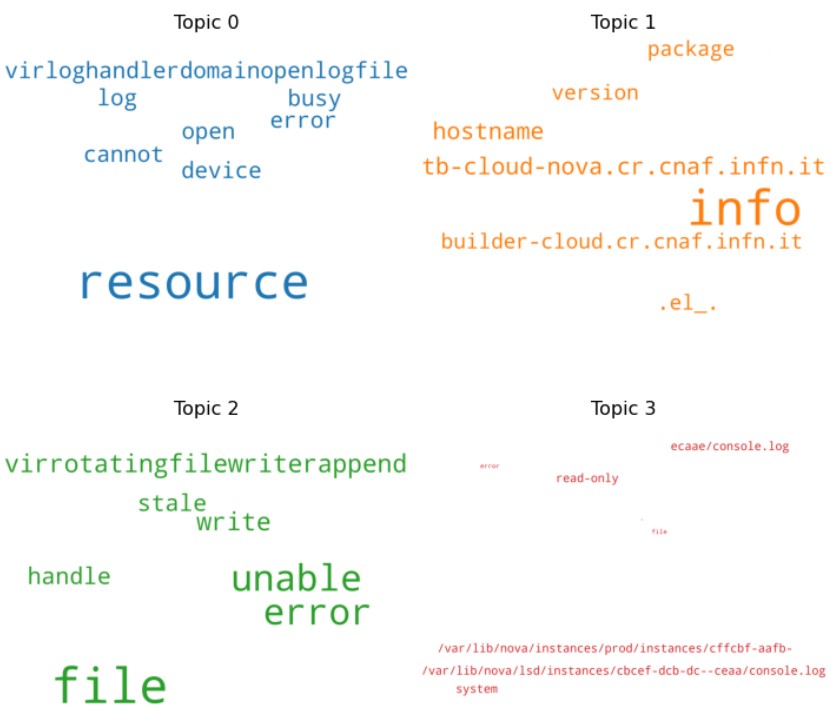

**Figure 10.** Identified topics for the *virtlogd* service.

*5.3. Feature Matrix*

Once the dictionary has been built, for each log file a feature matrix will be generated. By inserting such dictionary among the parameters of the TF-IDF function or more simply of a Count Vectorizer, it is possible to have a feature matrix strictly connected with the semantic area of the anomalies and able to detect if any log entry contains these patterns or not. The matrix will have as many rows as there are log messages and as many columns as there are elements that make up the dictionary (see Table 4). The elements of the array, on the other hand, indicate the number of times that the element y of the dictionary is contained in the message x. The matrix thus created has three main advantages: it is strictly connected to what was observed anomalous in that particular service; the matrix has a reduced dimension as it consider a smaller number of *n*-grams (i.e., *n* lower than 6) and is able to trace other parts of the message that better explain the type of anomaly. We again used CountVectorizer() to perform vectorization.

**Table 4.** Examples of message strings split up in single words for the crond log file.

| Log Event msg | .. | Error | .. | Failed | .. | Connection Time | .. |
|---|---|---|---|---|---|---|---|
| .. reset error counters | .. | 1 | .. | 0 | .. | 0 | .. |
| .. failed create session connection time out | .. | 0 | .. | 1 | .. | 1 | .. |

*5.4. Clustering Algorithm*

Each row of the feature matrix corresponds to a different log message. From how the feature matrix was built, each message is associated with a row vector, which contains count values. Consequently, starting from the feature matrix, those messages that had equal row vectors were grouped. An ad hoc clustering algorithm was built that would group log entries that had equal rows in the feature matrix. Lines that contain null values mean that no anomalous patterns have been found for that particular log message. These messages were grouped in 'Cluster 0'. Each cluster, excluding the Cluster 0, constitutes an anomalous message prototype common to one or more log entries.

Figures 11 and 12 show key terms, included in the dictionary, which constitute such prototypes for the *screen* and *virtlogd* services.

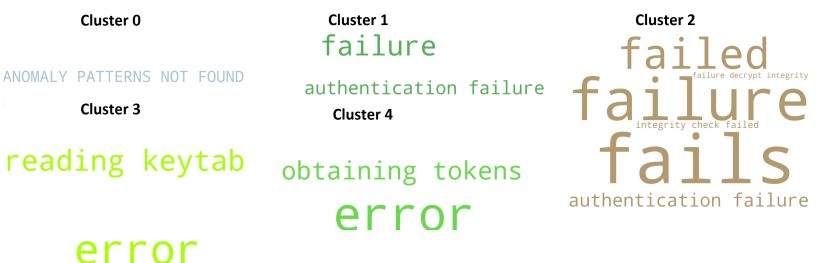

**Figure 11.** Grouping messages for the *screen* service.

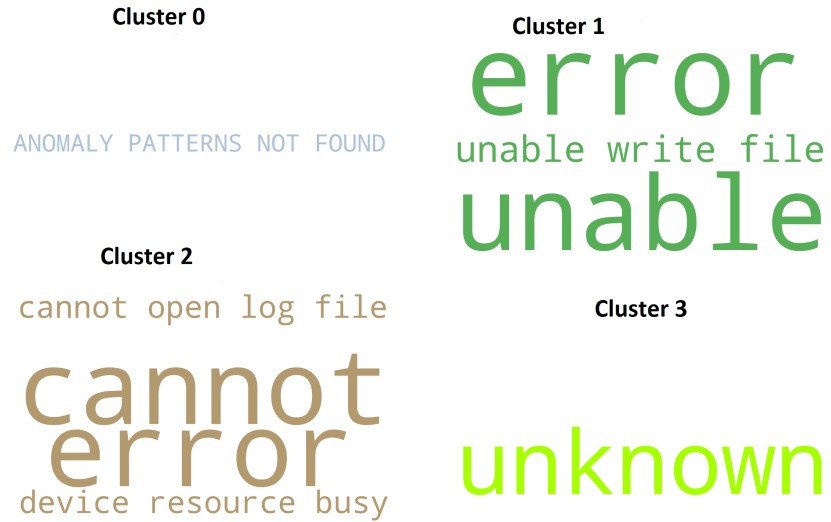

**Figure 12.** Grouping messages for the *virtlogd* service.

In Table 5 we report some log messages included in the *screen* log file. In the messages, "username" was written in place of the real usernames for privacy reasons. This table is useful to understand how our methodology was actually useful to identify anomalous and non-anomalous messages and to distinguish the different types. It is possible to notice the correspondence of the labels with the wordclouds in the Figure 11, which reports only the *n*-grams associated with the semantic area of anomalies and consequently included in the dictionary.

Through this clustering algorithm it is possible to identify anomalous messages, excluding those in 'Cluster 0'. In addition, the various types of anomalous messages were grouped into different clusters. According to the words and sequences of words identified as anomalies with the usage of NLP techniques, we have counted almost 499,110 thousand anomalies in all the log files over 3620 thousand observations. Once a new observation occurs, simply transform it into a count vector in the same way the feature matrix is created and compare it with the other rows in order to assign it a label. Although the dictionary creation part requires time and exploration of the text, the result is extremely reliable and adaptable to all types of log files. Also, a peculiarity of data center services is that they change over time. A supervised approach not only requires the parameters to be optimized and the model trained for each log file, but this process should be iterated with each change to the service. On the contrary, our unsupervised approach is more flexible and it would be enough to include further elements in the dictionary thanks to the help of the programmers who manage the changes in the logs. Thanks to the methodology used, it was possible to answer affirmatively to RQ1. Regarding RQ2, we believe that at the moment there are

no NLP techniques that automatically manage to understand if certain words or word sequences identify an anomaly in the particular context and without having any label. For this reason, a minimal human contribution is necessary and we believe that building a dictionary and our methodology are good contributions on how to proceed.

**Table 5.** Some log messages of the *screen* service and their cluster membership.

| Date | Time | Hostname | Process_Name | msg | Cluster |
|---|---|---|---|---|---|
| 21 January 2021 | 09:12:53 | ui-tier1 | screen | pam_krb5[19197]: TGT verified | 0 |
| 21 January 2021 | 09:12:53 | ui-tier1 | screen | pam_krb5[21445]: got error -1 (Unknown code ____ 255) while obtaining tokens for infn.it | 4 |
| 21 January 2021 | 09:12:53 | ui-tier1 | screen | pam_krb5[19197]: authentication succeeds for 'username' (username@CR.CNAF.INFN.IT) | 0 |
| 6 August 2020 | 12:27:22 | ui02-virgo | screen | pam_unix(screen:auth): authentication failure | 1 |
| 6 August 2020 | 12:27:22 | ui02-virgo | screen | pam_krb5[24018]: authentication fails for 'username' (username@CR.CNAF.INFN.IT): Authentication failure (Decrypt integrity check failed) | 2 |
| 6 August 2020 | 12:27:22 | ui02-virgo | screen | pam_ldap(screen:auth): Authentication failure | 1 |
| 6 August 2020 | 12:27:34 | ui02-virgo | screen | pam_krb5[24018]: authentication fails for 'username' (username@CR.CNAF.INFN.IT): Authentication failure (Decrypt integrity check failed) | 2 |
| 6 August 2020 | 12:27:34 | ui02-virgo | screen | pam_ldap(screen:auth): Authentication failure | 1 |
| 6 August 2020 | 12:27:50 | ui02-virgo | screen | pam_krb5[24018]: error reading keytab 'FILE:/etc/krb5.keytab' | 3 |
| 6 August 2020 | 12:27:50 | ui02-virgo | screen | pam_krb5[25026]: got error -1 (Unknown code ____ 255) while obtaining tokens for infn.it | 4 |

## 6. Anomaly Detection on Monitoring Metrics

The monitoring files are created with multiple headings, therefore we have removed some rows from the files before performing any analysis.

Some metrics show a noisy behavior, making the graphs unreadable and difficult to extract essential information about trends and large-scale deviations. At this moment of the study we have not applied any smooth functions to better identify trends in the metrics.

Figure 13 shows load average metrics at 1 min, 5 min and 15 min for a certain machine over time. The trends of the three metrics are very similar. The grey rectangle in the bottom plot is zoomed in the top plot. The plot on the top shows load average values below 0.1: they are between 0 and 0.02 except the end of December 2020 and the end of March 2021 where we observe an increase of 33% due to the usage of core in the machine. However, the measured values for these three metrics do not concern us.

*Anomaly Scores Resulting from JumpStarter*

Concerning the monitoring data, we have applied the JumpStarter solution [3], a multivariate time series anomaly detection based on compressed sensing to determine anomaly score and use it to label observations. This methodology is also completely unsupervised and calculates the anomaly score for each time instance, by comparing the original time series with the one reconstructed by the model. JumpStarter requires an input data matrix that has the metrics as columns and the instants in time in which the values were recorded by row. Since our monitoring observations were all recorded every hour of the analyzed period, it was not a problem to create such a matrix. Figure 14 shows the JumpStarter anomaly score resulting from memory metrics. In the same plot we have reported with vertical lines the anomalies checked in four log files for their corresponding services, such as *audit*, *haproxy*, *journal* and *sensu-service*. The plot on the top shows the anomaly score in the zoomed period where anomalies in the logs were found. The bottom plot shows

the trend of the anomaly score throughout the entire analysis period. It is noted that the graph is flat almost throughout the period, except for some small variations, but that the score increases in the period in which errors occur in the logs.

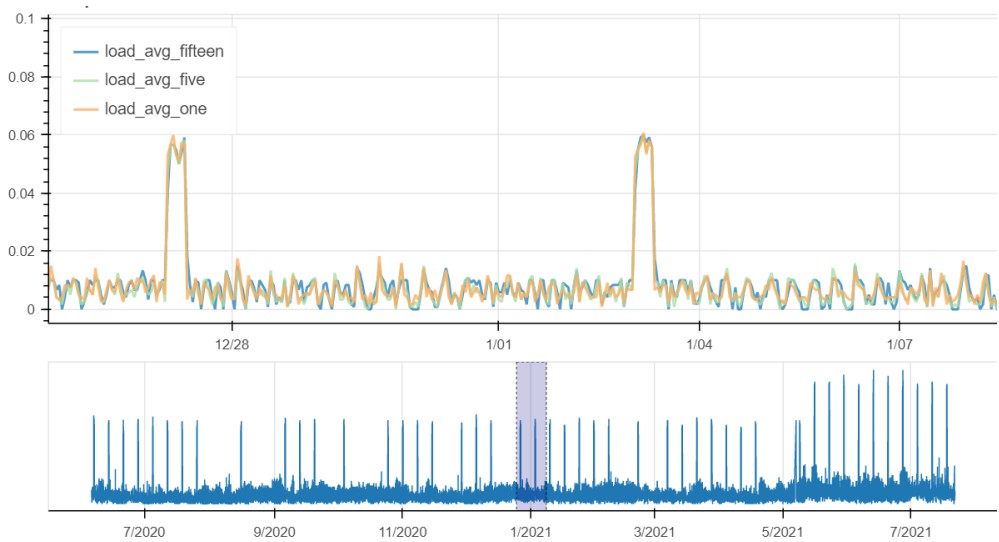

**Figure 13.** Comparison of load average metrics.

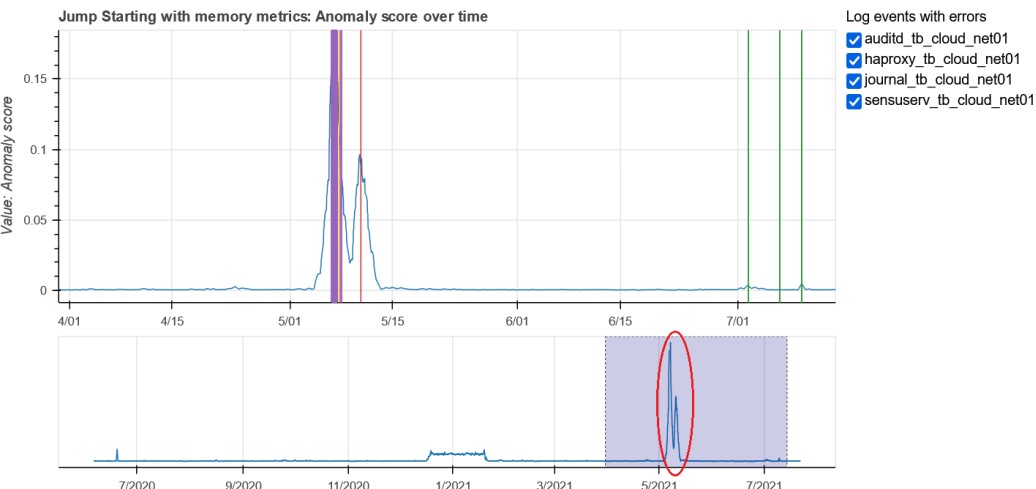

**Figure 14.** JumpStarter anomaly scores for the memory metrics on the *tb-cloud01-net* machine.

Figure 15 shows the JumpStarter anomaly score resulting from load metrics. Unlike the memory metrics, the trend of the anomaly scores for the load metrics is not flat but is characterized by fairly regular peaks just like the load metrics, see the Figure 16.

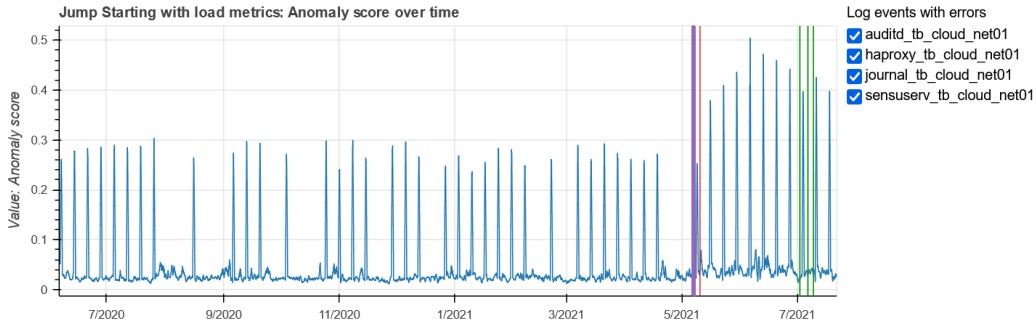

**Figure 15.** JumpStarter anomaly scores for the load metrics on the *tb-cloud01-net* machine.

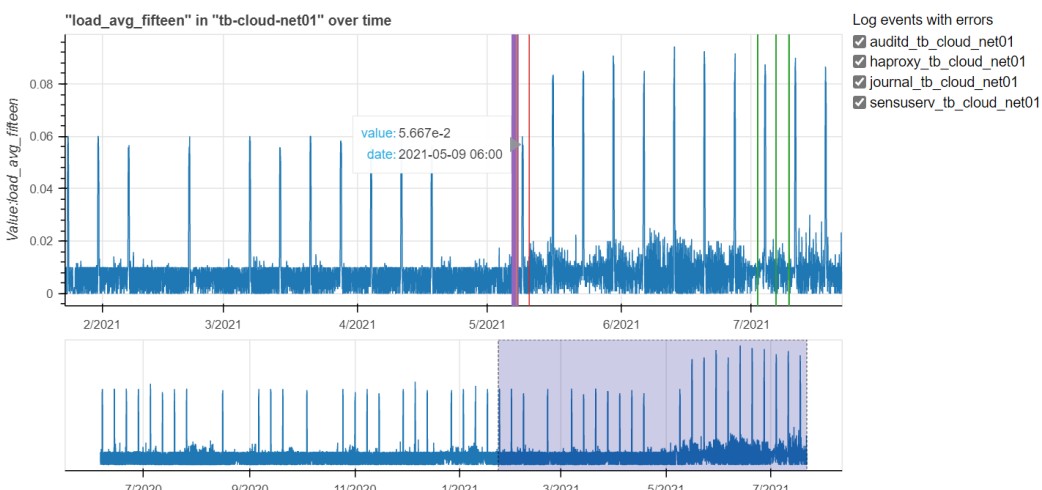

**Figure 16.** Overlapping log files and load averages at 15 min for the *tb-cloud01-net* machine.

The reason is that JumpStarter is a methodology that emphasizes sudden peaks in time series as anomalous, although these do not vary much from the "normal" situation. This methodology therefore requires more in-depth analysis by the experts. Future works could also think about smoothing out sudden peaks before applying the JumpStarter methodology and comparing it with other unsupervised methodologies.

## 7. Combining Anomalies at Machine Level

In this section, we are going to present results at machine level, combining what we have obtained during the log analysis and time series of monitoring data. We have produced a huge quantity of results, therefore we have decided to show what we consider the most suitable to explain our study. The considered machine is called *tb-cloud-net01* where the *audit*, *haproxy*, *journal* and *sensu-service* services run. The analyzed period goes from 6 June 2020 up to 21 July 2021.

Figures 16–18 show the time series of load, memory and iostat metrics respectively for the *tb-cloud-net01* machine, reporting anomalies of the considered services. The colored vertical lines represent anomalies identified in the log files of the selected services. Furthermore, each figure shows a grey rectangle in the bottom plot that is zoomed in the top plot.

Figure 16 shows load average metrics at 15 min. The plot on the top shows load average values below 0.1: they are mainly between 0 and 0.02, but there are some peaks due to the usage of cores in the machine. However, the values measured for these three metrics do not cause concerns.

Figure 17 shows the memory used average metric. The plot on the top shows the percentage of memory used compared to the total. The two plots highlight the variation in the metric after an intervention in one of the listed services, particularly a decrease in memory usage. The data center adopts the infrastructure policy to send a warning message when the metric's value excesses 95% of total memory as threshold. In this case this limit is not reached.

Figure 18 shows the iostat average for the cpu pct_iostat metric. The values do not cause concerns and the peaks are due to the cpu utilization.

The anomalies represented with the vertical lines in the previous figures are listed in Tables 6–8: they summarize what we have identified in the *auditd*, *journal* and *sensu-service* services respectively in the *tb-cloud-net01* machine from the 6 May 2021 at 15:00 to the 8 May 2021 at 03:00. In the same tables the cluster number each log message belongs to is reported.

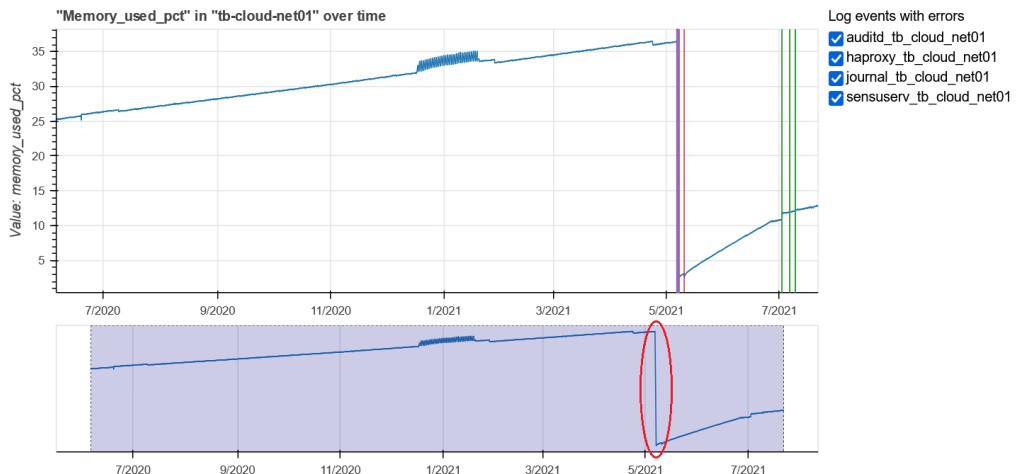

**Figure 17.** Overlapping log files and memory used values for the *tb-cloud01-net* machine.

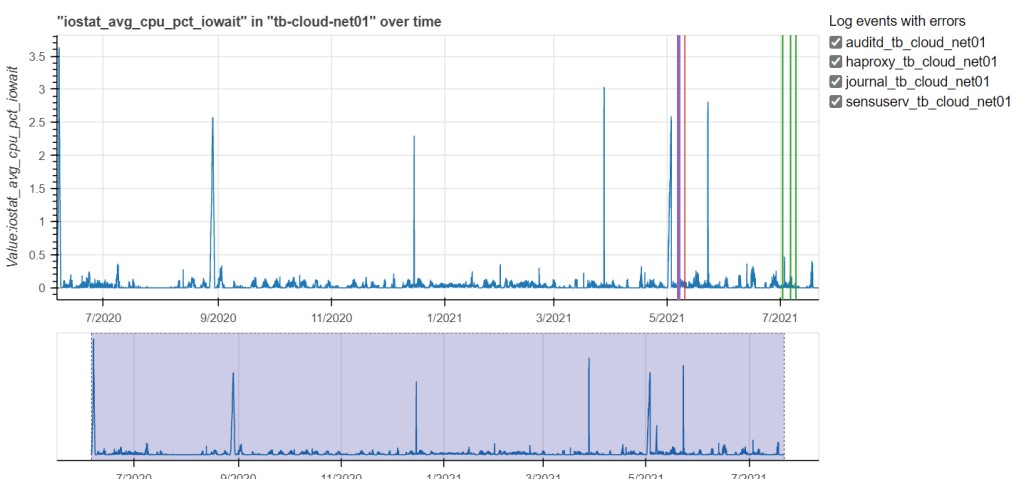

**Figure 18.** Overlapping log files and iostat average cpu pct_iowait values for the *tb-cloud01-net* machine.

**Table 6.** Anomalies recorded in *tb-cloud-net01* in the *auditd* service.

| Date | Time | Host_Name | Process_Name | msg | Cluster |
|---|---|---|---|---|---|
| 7 May 2021 | 17:33:50 | tb-cloud-net01 | auditd | Audit daemon is suspending logging due to previously mentioned write error | 1 |
| 6 May 2021 | 17:33:50 | tb-cloud-net01 | auditd | Audit daemon is suspending logging due to previously mentioned write error | 1 |

**Table 7.** Anomalies recorded in *tb-cloud-net01* in the *journal* service.

| Index | Date | Time | Host_Name | Process_Name | msg | Cluster |
|---|---|---|---|---|---|---|
| 0 | 7 May 2021 | 23:32:51 | tb-cloud-net01 | journal | Suppressed 18,739 messages from / | 2 |
| 1 | 7 May 2021 | 23:53:56 | tb-cloud-net01 | journal | Suppressed 5738 messages from / | 2 |
| 2 | 7 May 2021 | 23:53:56 | tb-cloud-net01 | journal | Suppressed 5672 messages from /system.slice/boot.mount | 2 |
| 3 | 7 May 2021 | 23:53:51 | tb-cloud-net01 | journal | Suppressed 19,279 messages from / | 2 |
| ... | | | | | | |
| 229 | 6 May 2021 | 23:32:26 | tb-cloud-net01 | journal | Suppressed 5640 messages from /system.slice/boot.mount | 2 |

**Table 8.** Anomalies recorded in *tb-cloud-net01* in the *sensu-service*.

| Index | Date | Time | Host_Name | Process_Name | msg | Cluster |
|---|---|---|---|---|---|---|
| 0 | 7 May 2021 | 23:54:30 | tb-cloud-net01 | sensu-service | {"level":"error","message":"log file is not writable", "log_file":"/var/log/sensu/sensu-client.log"} | 11 |
| 1 | 7 May 2021 | 23:54:30 | tb-cloud-net01 | sensu-service | {"level":"warn","message":"config file does not exist or is not readable", "file":"/etc/sensu/config.json"} | 12 |
| 2 | 7 May 2021 | 23:54:30 | tb-cloud-net01 | sensu-service | {"level":"warn","message":"ignoring config file", "file":"/etc/sensu/config.json"} | 13 |
| 3 | 7 May 2021 | 23:54:30 | tb-cloud-net01 | sensu-service | {"level":"warn","message":"loading config files from directory", "directory":"/etc/sensu/conf.d"} | 14 |
| 4 | 7 May 2021 | 23:54:30 | tb-cloud-net01 | sensu-service | {"level":"warn","message":"loading config file", "file":"/etc/sensu/conf.d/smart.json"} | 15 |
| ... | | | | | | |
| 2881 | 7 May 2021 | 02:53:44 | tb-cloud-net01 | sensu-service | {"level":"warn","message":"loading config file", "file":"/etc/sensu/conf.d/subscription_smartctl-os.json"} | 15 |
| 2882 | 7 May 2021 | 02:53:44 | tb-cloud-net01 | sensu-service | {"level":"warn","message":"config file applied changes", "file":"/etc/sensu/conf.d/subscription_smartctl-os.json","changes":{}} | 16 |

Answering to RQ3 and RQ4, not only is it possible to obtain the status of a machine by observing the monitoring metrics, but at the same time the anomalies found in the services provide additional information. Looking at the above tables with anomalies and the graph on the percentage of memory used, there is a correspondence between the occurrence of some anomalies in the logs and the sudden drop in the metric recorded in the period from 6 May 2021 to 7 May 2021. The percentage of memory used reached values below 5% in the same period in which anomalies, such as "suspending logging due to previously mentioned write error", "log file is not writable","config file does not exist or is not readable", occurred in the services. Extremely similar results were found for other machines. For this reason we suggest to combine the information obtained from the two data sources mentioned in this article.

## 8. Conclusions

In this paper we have discussed the specific findings from our study. The results include the data extraction, the exploration of server logs on physical and virtual resources, the investigation of monitoring metrics' data. They have been obtained applying natural language processing solutions on log files, clustering technique on log files and JumpStarter technique on monitoring data.

The data extraction has been challenging, having access to a huge amount of data. It has been hard to identify the proper use cases selecting log files and monitoring data on a given machine.

Up to now we have observed interesting outcomes thanks to the integration between log files and monitoring data. We have been able to map log anomalies into machines' metrics time series, noticing strange variation on monitoring data in the same period of time. Our model requires to integrate site administrators' expertise in order to consider all critical scenarios in the data center and understand results properly.

Furthermore, we also have to deeply investigate results obtained with the application of time series anomaly techniques, such as JumpStarter. A better understanding of the monitoring data may definitely boost this part.

With the collected information it will be easy to label both monitoring data and log data observation to predict anomalies.

**Author Contributions:** For Conceptualization, L.V. and E.R.; methodology, L.V. and E.R.; software, L.V. and E.R.; formal analysis, L.V.; investigation, L.V., E.R. and C.C.; resources, E.R.; data curation, L.V. and E.R.; writing—original draft preparation, E.R.; writing—review and editing, L.V., E.R. and C.C.; visualization, L.V., E.R. and C.C.; supervision, E.R.; project administration, E.R. All authors have read and agreed to the published version of the manuscript.

**Funding:** This work was partially supported by the EU H2020 IoTwins Innovation Action project (g.a. 857191).

**Institutional Review Board Statement:** Not applicable.

**Informed Consent Statement:** Informed consent was obtained from all subjects involved in the study.

**Data Availability Statement:** Not applicable.

**Acknowledgments:** The authors gratefully acknowledge INFN Tier-1 site administrators who provide log files.

**Conflicts of Interest:** The authors declare no conflict of interest.

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
