# Peer review of "Combining Log Files and Monitoring Data to Detect Anomaly Patterns in a Data Center"

_computers, doi:10.3390/computers11080117_

Round 1
Reviewer 1 Report
The submission discusses what seems to be a good idea; Analyze certain logfiles for some extreme events that seem to can be connected to eg. errors, failures etc. The problem with this submission is that it goes through a bunch of methodologies, without going into details neither on the methodologies nor the findings. Some examples: Topic modelling is used, I can clearly see the authors use different number of topics, but nothing is said on that. Topic modelling is a useful tool but needs a good deal of discussion on how the number of topics is selected, which implementation is followed and so on. Furthermore, on Figures 9, 10, 11 there are some spikes that were detected “for load averages at 15 minutes” (fig. 9) for the values of “iostat average cpu pct_iowait” (fig 11) . The results in both figures should definitely be something useful, but we don’t get this information from the paper. The journal audience would expect something that explains the events, when do those occur and what does it help for us to detect those occurrences.
Those are two examples of parts of the submission that need a more in-depth discussion but there are some more parts that need improvements.
The submission also needs a good proof reading, as there are some expressions that could be improved. A few examples, l. 86 "the rational of our approach"-> "the rationale of our approach", l. 209 is "abortion" really found in a logfile?!
I would therefore recommend to reject the paper and Ι would recommend the authors to provide more details in those subjects.
Author Response
C1: The submission discusses what seems to be a good idea; Analyze certain logfiles for some extreme events that seem to can be connected to eg. errors, failures etc. The problem with this submission is that it goes through a bunch of methodologies, without going into details neither on the methodologies nor the findings.
A1: We have improved the description of the methodology adopted: added a specific section 3 and changed Figure 1.
C2: Some examples: Topic modelling is used, I can clearly see the authors use different number of topics, but nothing is said on that. Topic modelling is a useful tool but needs a good deal of discussion on how the number of topics is selected, which implementation is followed and so on.
A2: We have improved the way we used topic modelling and selected the number of topics.
C3: Furthermore, on Figures 9, 10, 11 there are some spikes that were detected “for load averages at 15 minutes” (fig. 9) for the values of “iostat average cpu pct_iowait” (fig 11) . The results in both figures should definitely be something useful, but we don’t get this information from the paper.
A3: We have improved the description of each figures.
C4: The journal audience would expect something that explains the events, when do those occur and what does it help for us to detect those occurrences.
A4: We have tried to improve the description of the study: improved sections 3,4,5,6,7,8 and 9.
C5: Those are two examples of parts of the submission that need a more in-depth discussion but there are some more parts that need improvements.
A5: Improved sections 3,4,5,6,7,8 and 9.
C6: The submission also needs a good proof reading, as there are some expressions that could be improved. A few examples, l. 86 "the rational of our approach"-> "the rationale of our approach",
A6: Done
C7: l. 209 is "abortion" really found in a logfile?!
A7: We have checked all the selected log files. The word was put in the list wrongly. It has been removed.
C8: I would therefore recommend to reject the paper and Ι would recommend the authors to provide more details in those subjects.
A8: Please consider the revised version. Your feedback allow us to improve the previous version.
Reviewer 2 Report
This paper presents a method to detect anomaly patterns in a data center by combining log files and monitoring data. In general, the presented manuscript is interesting and generally well-written. However, there are still several issues that should be solved. The reviewer has the following comments:
1. The methodology part is not written in a clear manner. For example, what are the features used in anomaly detection? Which algorithm do the authors use in the anomaly prediction? More details should be added.
2. Please regenerate Figure 1 as there are some watermarks in the bottom left of Figure 1.
3. A minor suggestion is related to Figure 2. The current version actually contains a lot of unrelated information.
4. The word cloud shown in Figure 12 seems not to be the original one since there are many vague dots and trails. Please check it.
5. Besides, the image quality of some figures is relatively low (e.g., Figure 13). Please replace them with high-quality ones.
6. It is suggested to add some discussions about the failure cases.
Author Response
C1. The methodology part is not written in a clear manner. For example, what are the features used in anomaly detection? Which algorithm do the authors use in the anomaly prediction? More details should be added.
A1: Rewritten section 3, improved section 5 about features creation and section 7 about ML techniques.
C2. Please regenerate Figure 1 as there are some watermarks in the bottom left of Figure 1.
A2: Done
C3. A minor suggestion is related to Figure 2. The current version actually contains a lot of unrelated information.
A3: Added a better description of Figure 2. We consider it important because it summarizes the categories of resources at INFN Tier 1 for which we have collected data.
C4. The word cloud shown in Figure 12 seems not to be the original one since there are many vague dots and trails. Please check it.
A4. Improved the Figure where info about users have been removed.
C5. Besides, the image quality of some figures is relatively low (e.g., Figure 13). Please replace them with high-quality ones.
A5. Done
C6. It is suggested to add some discussions about the failure cases.
A6. Added in section 8
Round 2
Reviewer 1 Report
The article is significantly revised, but it has a series of issues that I am trying to highlight below. I can recommend accepting the article after considering the points I am posting below. A general comment is that the authors need to clarify their methodology in some points.
· L 79-80 “For anomaly detection, however, it may be necessary to find rare types of messages and interesting patterns, so word n-grams solution has been also applied.”
- How word n-grams helps with that?
· L. 95 “putting into relation” -> “relate”. The same expression is used in line 173.
· L. 97 the reminder -> the remainder
· l. 81-l.85
The clustering methods in the log core analysis phase would lead to the risk of losing significant results on the anomalies to be considered individually: for this reason we have used topic modelling techniques.
Clustering loses some information on the outliers. Reference [5] seems to alleviate this problem by clustering the outliers. Topic modelling (eg. LDA) can also have some issues with rare events, is this solved by using n-gram methods? Please discuss.
They also use n-grams to decide on the less frequent ones?
· l. 176-192: Methodology overview -
The Methodology section has been added to the submission and it is a much-needed addition. I would suggest the authors to be more specific with the techniques they used, naming the techniques here and giving the appropriate references to them. By saying "data preprocessing", "NLP techniques" and "time series anomaly detection" you are covering 60% of data science!
The authors discuss very briefly about word2vec in the beginning of section 7, giving a reference there (!) to their work. Reference to [2] is fine but one would rather here some more detail in a journal version of this paper, like your current submission. Some questions rising are: Why are you specifically using word2vec? Which feature(s) of this technique make you use it?
Later in your submission you also discuss about Topic Modelling. Make some introduction here, with a reference to the exact topic-modelling technique you are using.
· l.194 “the datasets selection” -> dataset selection
· l. 205-206: it seems that the authors use basically only the *.log files. This is still a bit vague when we read the paper, please justify.
· l. 207-208: Please clarify if you are only using .log files. That's what I understand by reading your work but you mention a variety of other file types. Are those all logfiles? I am not sure what .gz and .txt can contain.
· l. 272: the service name included -> the service name is included. Same with the process identifier.
· L. 263-264: the authors write that they have changed to format of logfiles to CSV? Do you use the space character as field separator? That seems to me to be the case, because the fields like the ones shown in figures 3 to 6, date, time, hostname, are said to be the “variables that have been added”.
· L. 269 “location is got by a local database” –> “location is obtained by a local database”.
· L. 279-282: There is nothing called “cancellation of stopwords”. Better change it to “stopword removal”.
· L. 286 “contributing to better understand variation” -> “contributing to a better understanding of the variation”
· l. 291: so isn’t “abortion” removed from the log messages? I wrote this in the previous review, too, so if it is removed from the log messages, please remove from here, too.
· Section 5. Features transformation (please change it to “Feature transformation”)
There are some things that cause confusion here:
You are writing: “For this reason, for each log file it is necessary to reduce the dimensionality of the features matrix, by restricting the number of columns to n-grams that are related to any problems or warnings.”
In other words the “terms” in the TF-IDF matrix are substituted by the n-grams, is that right? You then write that it is necessary to create a dictionary that contains words or n-grams related to problems. What is exactly the structure of such a dictionary? Can you give an example? How is such a dictionary constructed? When you say: “In addition to the help from data center experts, it might be useful to explore the complete list of n-grams starting from the texts” you mean that you combine experts and n-grams?
Then do you check the most frequent n-grams and then add them to a dictionary? Is it just a matter of frequency, or can other metrics like t-score be used? Give a concrete example of how you are constructing this dictionary and some part of the dictionary’s contents as well as how the Count Vectorizer is involved.
My guess is that you are processing the corpus of log files and metrics’ files, you are isolating n-grams indicated by your dictionary, give those n-grams as input to the CountVectorizer and then apply a dimensionality reduction technique, like word2vec (which is not only dimensionality reduction but this is how it gets used in the paper). If this corresponds to the truth, please outline your methodology appropriately in the paper. A good place would be the section “Methodology Overview” but some ideas can also be written in Section 5.
· L. 323-333 Topic modelling:
There are many observations in this section. I would also recommend to write this part in a separate subsection.
Which exactly method are you using. Is it Latent Dirichlet Allocation, Non-negative Matrix Factorization or something else?
You have included the text documenting selection of the number of topics, which now makes more sense. All of the text in lines 323-333, makes very good sense. What does not make very much sense is how Topic Modelling is used! Topic Modelling helps you find latent and prevalent topic, that’s true. How are those topics used at the end? My guess is that you are going through the topics (perhaps manually?), identify topics that correspond to errors and are then able to highlight some documents that seem to describe error. Which then brings me to another question; When you apply topic modelling do you still consider every line of the log files, as a separate document? Please discuss.
A last observation; I have seen a kind of preprocessing before applying topic modelling, where terms are grouped as n-grams and then each n-gram is used as a “term” in the Topic Modelling. This could lead to something like eg. the Topic 0 in Figure 9, to have terms like “cannot_open_device” as one word, because right now it looks as if you have three separate, but very closely related words.
· Lines 334-336, those lines seem to be related to the TF-IDF. Otherwise are you first preprocessing with a dictionary and then TF-IDF and then Topic Modelling. Please comment
· 7. Machine Learning Techniques:
Firstly, this title is a slight misnomer. Topic Modeling is also Machine Learning… Then the authors write “In this work, we have defined a clustering algorithm in order to group log messages”. Which algorithm did you define and where? This looks superficially like Topic Modelling. Subsequently, the paper goes very quickly through the JumpStarter solution. This solution is not discussed at the submission, which does not help in understanding the solution plus the results are very quickly presented without comments. The authors further state “The results of this technique requires a further investigation.”. I would recommend excluding this section from the submission.
· Line 387: “at 15 minutes” -> “every 15 minutes”
· L. 388: “picks” -> “peaks”. The same typo on line 398
· L. 390: “the values are not worrying” -> “the values do not cause concerns”. This appears in other parts of the text, too.
· L. 394: “to send an alarming message” -> “to send a warning message”.
· Tables 5-7: I would recommend you to comment on the number of clusters. It seems that clustering only makes sense for the sensu service. Are you applying the clustering from Section 7?
Author Response
Dear Reviewer,
in the following you can find our replies to your questions. We have reorganized sections 3, 4, 5, 6, 7 in order to cover your recommendations.
R0 A general comment is that the authors need to clarify their methodology in some points.
A0 Improve section 4
R1 L 79-80 “For anomaly detection, however, it may be necessary to find rare types of messages and interesting patterns, so word n-grams solution has been also applied.” How word n-grams helps with that?
A1 We have considered the list of words and sequence of words identified with n-grams. We have avoided to consider just split terms avoiding the meaning of each messages.
R2 L. 95 “putting into relation” -> “relate”. The same expression is used in line 173.
A2 Done
R3 L. 97 the reminder -> the remainder
A3 Done
R4 l. 81-l.85 “The clustering methods in the log core analysis phase would lead to the risk of losing significant results on the anomalies to be considered individually: for this reason we have used topic modelling techniques.”
“Clustering loses some information on the outliers. Reference [5] seems to alleviate this problem by clustering the outliers. “ Topic modelling (eg. LDA) can also have some issues with rare events, is this solved by using n-gram methods? Please discuss.
A4 Improved section 5
R5 They also use n-grams to decide on the less frequent ones?
A5 Please read section 5
R6 l. 176-192: Methodology overview - The Methodology section has been added to the submission and it is a much-needed addition. I would suggest the authors to be more specific with the techniques they used, naming the techniques here and giving the appropriate references to them. By saying "data preprocessing", "NLP techniques" and "time series anomaly detection" you are covering 60% of data science!
The authors discuss very briefly about word2vec in the beginning of section 7, giving a reference there (!) to their work. Reference to [2] is fine but one would rather here some more detail in a journal version of this paper, like your current submission. Some questions rising are: Why are you specifically using word2vec? Which feature(s) of this technique make you use it?
Later in your submission you also discuss about Topic Modelling. Make some introduction here, with a reference to the exact topic-modelling technique you are using.
A6
R7 l.194 “the datasets selection” -> dataset selection
A7 Done
R8 l. 205-206: it seems that the authors use basically only the *.log files. This is still a bit vague when we read the paper, please justify.
A8 Modified this part in the following way:
The overall log suffix-type frequency is $3,562,759$. However, on the Linux machine other suffix-types of files (containing log entries) are available, such as \textit{.gz} and \textit{.txt} respectively with $10,869$ and $2$ frequencies. The \textit{.gz} files are used for large log files, while the \textit{.txt} files are for small log files.
R9 l. 207-208: Please clarify if you are only using .log files. That's what I understand by reading your work but you mention a variety of other file types. Are those all logfiles? I am not sure what .gz and .txt can contain.
A9 See previous answer. Put in the previous paragraph the following text:
In this work we have considered the same set of log files used in the previous study~\cite{ICCSA_2021}.
R10 l. 272: the service name included -> the service name is included. Same with the process identifier.
A10 Done
R11 L. 263-264: the authors write that they have changed to format of logfiles to CSV? Do you use the space character as field separator? That seems to me to be the case, because the fields like the ones shown in figures 3 to 6, date, time, hostname, are said to be the “variables that have been added”.
A11 Well not exactly. We have defined procedures to consider different file structure. The whole message has been considered as a string. Each log file has been changed in CSV according to specific-service procedure. We have identified services with the same log structure. Modified the first sentence in Data Preprocessing as follows:
During the data preprocessing we have first changed the format of log files, which have turned into .csv files by applying specific-service procedure.
R12 L. 269 “location is got by a local database” –> “location is obtained by a local database”.
A12 Done
R13 L. 279-282: There is nothing called “cancellation of stopwords”. Better change it to “stopword removal”.
A13 Done
R14 L. 286 “contributing to better understand variation” -> “contributing to a better understanding of the variation”
A14 Done
R15 l. 291: so isn’t “abortion” removed from the log messages? I wrote this in the previous review, too, so if it is removed from the log messages, please remove from here, too.
A15 Done
R16 Section 5. Features transformation (please change it to “Feature transformation”)
A16 This section has been reorganized in Section 5.
R17 There are some things that cause confusion here:
You are writing: “For this reason, for each log file it is necessary to reduce the dimensionality of the features matrix, by restricting the number of columns to n-grams that are related to any problems or warnings.”
In other words the “terms” in the TF-IDF matrix are substituted by the n-grams, is that right? You then write that it is necessary to create a dictionary that contains words or n-grams related to problems. What is exactly the structure of such a dictionary? Can you give an example? How is such a dictionary constructed? When you say: “In addition to the help from data center experts, it might be useful to explore the complete list of n-grams starting from the texts” you mean that you combine experts and n-grams?
Then do you check the most frequent n-grams and then add them to a dictionary? Is it just a matter of frequency, or can other metrics like t-score be used? Give a concrete example of how you are constructing this dictionary and some part of the dictionary’s contents as well as how the Count Vectorizer is involved.
My guess is that you are processing the corpus of log files and metrics’ files, you are isolating n-grams indicated by your dictionary, give those n-grams as input to the CountVectorizer and then apply a dimensionality reduction technique, like word2vec (which is not only dimensionality reduction but this is how it gets used in the paper). If this corresponds to the truth, please outline your methodology appropriately in the paper. A good place would be the section “Methodology Overview” but some ideas can also be written in Section 5.
A17 Please read section 5
R18 L. 323-333 Topic modelling:
There are many observations in this section. I would also recommend to write this part in a separate subsection.
Which exactly method are you using. Is it Latent Dirichlet Allocation, Non-negative Matrix Factorization or something else?
You have included the text documenting selection of the number of topics, which now makes more sense. All of the text in lines 323-333, makes very good sense. What does not make very much sense is how Topic Modelling is used! Topic Modelling helps you find latent and prevalent topic, that’s true. How are those topics used at the end? My guess is that you are going through the topics (perhaps manually?), identify topics that correspond to errors and are then able to highlight some documents that seem to describe error. Which then brings me to another question; When you apply topic modelling do you still consider every line of the log files, as a separate document? Please discuss.
A last observation; I have seen a kind of preprocessing before applying topic modelling, where terms are grouped as n-grams and then each n-gram is used as a “term” in the Topic Modelling. This could lead to something like eg. the Topic 0 in Figure 9, to have terms like “cannot_open_device” as one word, because right now it looks as if you have three separate, but very closely related words.
A18 Improved section 5
R19 Lines 334-336, those lines seem to be related to the TF-IDF. Otherwise are you first preprocessing with a dictionary and then TF-IDF and then Topic Modelling. Please comment
A19 Improved section 5
R20· 7. Machine Learning Techniques:
Firstly, this title is a slight misnomer. Topic Modeling is also Machine Learning… Then the authors write “In this work, we have defined a clustering algorithm in order to group log messages”. Which algorithm did you define and where? This looks superficially like Topic Modelling. Subsequently, the paper goes very quickly through the JumpStarter solution. This solution is not discussed at the submission, which does not help in understanding the solution plus the results are very quickly presented without comments. The authors further state “The results of this technique requires a further investigation.”. I would recommend excluding this section from the submission.
A20 We consider useful this part to introduce the fact that we can combine results from monitoring and log files.
R21 Line 387: “at 15 minutes” -> “every 15 minutes”
A21 Done
R21 L. 388: “picks” -> “peaks”. The same typo on line 398
A21 Done
R22 L. 390: “the values are not worrying” -> “the values do not cause concerns”. This appears in other parts of the text, too.
A22 Done
R23 L. 394: “to send an alarming message” -> “to send a warning message”.
A23 Done
R24 Tables 5-7: I would recommend you to comment on the number of clusters. It seems that clustering only makes sense for the sensu service. Are you applying the clustering from Section 7?
A24 Please read section 7
Reviewer 2 Report
The authors have addressed all my concerns. I have no more comments.
Author Response
Dear Reviewer
thank you for your feedback.
Round 3
Author Response
Reply to reviewers' comments.
This is a much-improved version and now the whole procedure has a specific flow and its parts are sufficiently connected with each other. Below you can find some more specific comments.
R1 L. 84 add here reference [13] to JumpStarter
A1 Done
R2 L. 203 “its value excesses 95%” -> “its value exceeds”
A2 Done
R3 Log files to CSV, Section 5.1: The authors note that log files are converted to CSV by following a “specific-service procedure”. First of all, if you mean a procedure depending on what service is been used, then you would better call it “service-specific”. Furthermore, it is not very clear, what the relevant procedure is. In their reply, the authors write “The whole message has been considered as a string”. Still it is very unclear, how you turn those messages into CSV. I would suggest that you give an example converting messages in eg. Figure 2 or Figure 3, into a CSV.
A3 Saying that the format of files is turned into csv files by applying service-specific procedure, means that we have had to consider different log header information. So this service-specific procedure is able to manage each entry of the file, e.g. adding e.g. missing information like IP address (as shown in Figure 2) or hostname (as shown in Figure 3), getting service name or process ID from the log message and adding it to the entry, and so on. The entry in the CSV file is not just the log message, it also contains timestamp, host, ip, service name, process id, level and log message.
We have tried to improve our description in 3.1
The log header structure is relatively different in both Figures: this is one of the reasons why the application of a non service-specific solution when parsing log files is a hard task.
Also in 5.1.
During the data preprocessing we have first changed the format of log files, which have turned into \textit{.csv} files by applying service-specific procedure, because we have had to take into consideration different log header structures (as shown in Figures~\ref{fig:ls1} and ~\ref{fig:ls2}), one per service. This procedure is able to manage each entry of the log file, performing at least the following operations: adding e.g. either internet protocol (ip) address or hostname when they are missing, service name when it is missing; splitting service name and process identifier (id); and getting component name. When the log file is turned into \textit{.csv} format, the entry in the \textit{.csv} file contains the log message (msg) and a set of other variables, such as: date, time, timestamp, hostname, ip address, service name, id, component name. The hostname and ip couples are not always both available, especially when the service runs on a virtual machine. Each file is related to a particular service that runs on a well-known machine in the data center. Its location is obtained by a local database and included into the resulting file.
R4 L. 285, “non-alphabets”: You probably mean non-alphanumeric characters.
A4 Done
- Section 5.2
R5 o In the beginning of Section 5.2, TF-IDF is explained and it is discussed about the “words” that appear in TF-IDF, which is what one would expect in the baseline scenario. The authors say in line 308 “since each unique n-gram in all messages will compose a column of such matrix.”. This makes us understand that you are using ngrams as words, which is a bit different than what a book like Manning, Raghavan and Schütze, Introduction to Information Retrieval, Cambridge University Press, 2008, suggests. This makes sense, and it is supported by libraries like scikit-learn but has pros and cons.
Please be more specific on the n-grams you chose. How do you decide on the value of n? This is also relevant to Figure 7, right? It is almost obvious that the n-gram we are searching for is “audit demon suspended logging due previous mentioned write error”. Does it make sense to break it into the 3-grams appearing in right part of Figure.7? Is this how you implement the n-gram extraction, at the end?
A5 Each log file can contain similar messages that include anomalous pattern or terms useful to differentiate anomalous messages. Furthermore, each message is composed by various dynamic fields. According to these messages characteristics, we have used several n-grams with n <=5 to rebuild anomalous message omitting dynamic fields. The n value selection depends on messages included in the log files.
R6 o You say in line 312, that “for each log file it makes sense to build a different dictionary”. There is a similar statement on line 76. Does this mean that you are building a dictionary for each log file, ending up with 3.500.000+ dictionaries? My understanding is that for each log file you have collected, you run the processing and add the relevant terms to a global dictionary. Please confirm your method.
A6 Your understanding is correct. I have added the following text:
For each log files it makes sense to build a different dictionary as the message structure and the type of anomalies are different between services. Doing so, we have been able to identify relevant terms to a global dictionary. The current procedure requires human intervention, therefore the procedure to build the dictionary can be further improved.
R7 o You are discussing about the most frequent n-grams and least frequent n-grams. Why do you chose the least frequent? What happens in the middle, with the ngrams of medium frequency? Are those n-grams ignored?
A7 We have used the whole list of terms and sequences provided by n-grams. We have selected the least frequent, mainly because a service is built to work, therefore anomaly terms have a lower frequency. N-grams of medium frequency have also been considered.
We have decided to consider n belongs to [3,5] because they are able to rebuild the meaning of each message better than with n equal to 2.
For simple and small log file, whenever possible we have performed a first handson check of the list of unique n-grams, and added them to anomaly dictionary.
R8 o When it comes to LDA it is now clear that you use the topics to enrich the dictionary with execution anomalies. You also say in line 348 “We have included anomaly pattern resulting from LDA, in the dictionary”. This needs some discussion! The topics seem to contain single words, some path-like names and domain names. Now, when there is a discussion on adding topics for the screen service, we see on Figure 8-topic 3, words like “auth”, “failure”, “authentication”, “screen”, “pam_ldap”. From a fast interpretation, this topic can indicate “screen authentication failure”.
A8 Modified text as follows:
In both Figures we can observe that during the preprocessing activity we have cleaned log messages keeping meaningful information, such as the function name, the name of the machine, users' names (that for privacy reason we have omitted), and potential problems. For example, Figure~\ref{fig:st}-Topic 2 shows words like \textit{auth}, \textit{failure}, \textit{authentication}, \textit{screen} and \textit{pam\_ldap}, that can indicate a \textit{screen authentication failure}. We have included in the dictionary construction, anomaly patterns resulting from LDA.
R9 â–ª Which of those terms do you include in the dictionary? Do you think it would be better to preprocess the logfiles, eg. extract ngrams and then feed them to LDA, so that you don’t have separate terms appearing like in Topic 2?
A9 For example, according to Figure~\ref{fig:st}, we have definitely added words \textit{failure} and \textit{authentication failure} from Topic 2; \textit{fails}, \textit{failed authentication} from Topic 0; \textit{unknown} from Topic 3. Instead we have omitted Topic 1 that contains \textit{verified} and \textit{authentication} that can indicate no problem.
LDA has been used to explore deeply each message and see if it was easy to identify topics.
In this study we have considered single terms for topic modeling due to the service specific knowledge included in each message (i.e. url, process name and so on).
R10 â–ª Are you leaving out topics like Topic 1? Terms like “authentication”, “verified”, are an indication that there are no errors, right? I would suggest that you discuss, which topics you include and how have you decide which to use and which not.
A10 For example, according to Figure~\ref{fig:st}, we have definitely added words \textit{failure} and \textit{authentication failure} from Topic 2; \textit{fails}, \textit{failed authentication} from Topic 0; \textit{unknown} from Topic 3. Instead we have omitted Topic 1 that contains \textit{verified} and \textit{authentication} that can indicate no problem.
R11 â–ª I think it is better to change “We have included anomaly pattern resulting from LDA, in the dictionary”, with “We have included in the dictionary construction, anomaly patterns resulting from LDA.”.
A11 Done
R12 â–ª Can you give an example of a latent topic that you would not have discovered, if you did not use LDA?
A12 We do not identify any latent topic.
- Section 5.3
R13 o As you are using CountVectorizer(), this now leads us to the scikit-learn library right? This now open a part that should be improved in your submission; As you begin to discuss very specific technical details, you need to write a small section, discussing on your implementation. I would suggest that you add a subsection “4.2 Project implementation”, or similar, where you discuss platform (Python?), libraries (scikitlearn, what is used for LDA, etc.) and cite your repository, if this is possible.
A13 STO RECUPERANDO LE INFO DAL CODICE (ELI)
R14 o If you discuss about the implementation, you could probably insert a comment on the n-gram generation, that connects to sklearn, since sklearn offers this method to generate n-grams for a range of values for n.
A14 Added (i.e. n lower than 6)
We also used CountVectorizer to generate n-gram.
CountVectorizer() has been used in two ways. Firstly, we use it to determine the whole list of unique n-grams in every log gile. Secondly, we use CountVectorizer() to build the feature matrix by considering n-grams included in the dictionary: the number of times one n-gram value is in the message.
R15 o L. 361: “as they consider”. “they” refers to whom?
A15 Replace with it (the matrix)
R16 o Table 4. And line 367: The array columns are single words but you have prepared us so far that you are using n-grams. I would suggest that you use “connection time out” instead of time in Table 4.
A16 Done
R17 - Section 5.4: It is interesting to see more details on this clustering algorithm. I would recommend citing an article or technical report with the algorithm. Otherwise, the discussion is very blurry about an algorithm that groups equal rows. Applying the algorithm seems to be having some positive effects that’s why it is recommended to give the algorithm in some form.
A17 We agree but the cluster algorithm is going to be published in the master thesis, and we do not have yet a reference because it is not yet published.
R18 - L. 401 “multiple heading” -> Do you mean “multiple headings”?
A18 Done
R19 - L. 411 change “not worrying”!
A19 However, The measured values for these three metrics do not concern us.
As a closing remark, there are still some typos and grammatical errors so make sure you do a better proof reading